# Experimental Modeling of Host–Bacterial Interactions in Head and Neck Squamous Cell Carcinoma

**DOI:** 10.3390/cancers15245810

**Published:** 2023-12-12

**Authors:** Ogoegbunam Okolo, Emily Honzel, William R. Britton, Victoria X. Yu, Samuel Flashner, Cecilia Martin, Hiroshi Nakagawa, Anuraag S. Parikh

**Affiliations:** 1Herbert Irving Comprehensive Cancer Center, Columbia University, New York, NY 10027, USA; obo2108@cumc.columbia.edu (O.O.); wrb2120@cumc.columbia.edu (W.R.B.); vxy2101@cumc.columbia.edu (V.X.Y.); sf3070@cumc.columbia.edu (S.F.); cm4194@cumc.columbia.edu (C.M.); hn2360@cumc.columbia.edu (H.N.); 2Columbia Vagelos College of Physicians and Surgeons, Columbia University, New York, NY 10027, USA; ech2169@cumc.columbia.edu; 3Department of Otolaryngology-Head and Neck Surgery, Columbia University, New York, NY 10027, USA; 4Organoid and Cell Culture Core, Columbia University Digestive and Liver Diseases Research Center, Columbia University, New York, NY 10027, USA; 5Division of Digestive and Liver Diseases, Department of Medicine, Columbia University, New York, NY 10027, USA

**Keywords:** head and neck squamous cell carcinoma, microbiome, in vitro modeling, organotypic culture systems, 3D organoids

## Abstract

**Simple Summary:**

The microbiome has been implicated in both homeostasis and disease states, including cancer. Mechanistic investigations into the role of the microbiome in head and neck squamous cell carcinoma (HNSCC) are in their relative infancy. To date, the literature suggests an altered microbiome in subjects with HNSCC, but the mechanisms behind these changes remain to be elucidated. The use of in vitro models utilizing co-culture of cancer cells with microbes, including traditional monolayer culture, 3D organotypic culture, and 3D organoids, may help characterize the underpinnings of the complex relationship between the microbiome and HNSCC, with the goal of improving risk stratification and ultimately guiding treatment.

**Abstract:**

The microscopic species colonizing the human body, collectively referred to as the microbiome, play a crucial role in the maintenance of tissue homeostasis, immunity, and the development of disease. There is evidence to suggest associations between alterations in the microbiome and the development of head and neck squamous cell carcinomas (HNSCC). The use of two-dimensional (2D) modeling systems has made significant strides in uncovering the role of microbes in carcinogenesis; however, direct mechanistic links remain in their infancy. Patient-derived three-dimensional (3D) HNSCC organoid and organotypic models have recently been described. Compared to 2D models, 3D organoid culture systems effectively capture the genetic and epigenetic features of parent tissue in a patient-specific manner and may offer a more nuanced understanding of the role of host–microbe responses in carcinogenesis. This review provides a topical literature review assessing the current state of the field investigating the role of the microbiome in HNSCC; including in vivo and in vitro modeling methods that may be used to characterize microbiome–epithelial interactions.

## 1. Introduction

Head and neck squamous cell carcinoma (HNSCC) affects an estimated 890,000 new patients every year across the world [1]. Stratification of this patient population is difficult, due to the existence of few clinically validated biomarkers and a heterogenous patient population [2]. Clinicians often rely on clinicopathologic features to guide decision-making, including depth of invasion, lymphovascular invasion, and perineural invasion [3]. However, there remains debate regarding specific clinical implications of many of these metrics [2]. While outcomes have improved for human papillomavirus (HPV)-positive cancers, the prognosis of HPV-negative HNSCC remains largely stable [4]. There remains an urgent need to further understand the mechanistic underpinnings of the pathogenesis of HNSCC, and one component of this may be understanding the role of the microbiome.

The microbes colonizing the human aerodigestive tract, collectively referred to as the microbiome, have been increasingly investigated in recent years and have been implicated in many disease states, including HNSCC [5,6]. Emerging evidence suggests that the pathogenesis of HNSCC may be affected by ecological principles such as host–microbe interactions, cellular communication, and competition at the tumor–host interface to create multidimensional intertumoral and intratumoral heterogeneity [7]. However, there remains controversy regarding the specific microbiome profiles in HNSCC tissue implicated in cancer microecology, with a lack of consistency between studies potentially due to limitations in sample collection and modeling techniques.

The examination of host–microbe interactions in disease development warrants the use of in vitro tissue culture systems containing microorganisms to adequately model in vivo host biology. While two-dimensional (2D) modeling of host–microbe interactions has provided useful evidence linking candidate microbes with features such as epithelial invasion and proliferation [8,9], these modeling systems may be limited in their ability to recapitulate the multi-layer epithelial architecture present in human host tissue. Three-dimensional (3D) in vitro models such as organotypic and 3D organoid culture systems have been utilized to bridge this gap, serving as a relevant tool to examine microbe–tissue interactions in the setting of carcinogenesis [10]. While 3D culture systems hold significant potential for investigating host–microbe interactions in the context of oral cavity HNSCC, their utilization in this area is still in its nascent stages.

This topical review describes the bacterial microbiome of the head and neck, with particular attention to the altered microbiome present in subjects with HNSCC. Techniques used to investigate tumor–microbiome interactions are explored, with particular attention given to the utility of monolayer cell culture techniques and the potential of 3D culture techniques as novel tools for HNSCC tumor–microbiome profiling.

## 2. The Microbiome Plays a Role in Head and Neck Homeostasis

The human body is home to thousands of bacteria. The microbiome of the intestinal tract is perhaps best characterized, with recent attention given to the microbiota of the head and neck. A recent review of the literature highlighted the diverse bacterial profiles discovered in anatomic subsites within the head and neck, including the sinuses, larynx, and ear [11]. Furthermore, the oral microbiome, which itself consists of over 700 unique species of bacteria, can be divided into anatomic regions with unique bacterial communities, including the dorsum of the tongue, hard palate, soft palate, subgingival plaque, and others [12]. Although these anatomically distinct profiles throughout the aerodigestive tract have been defined, the sites are highly related and likely influence each other. For example, the administration of chlorhexidine mouthwashes modified the microbiomes of both the treated oral cavity as well as that of the distal esophagus, highlighting the interrelatedness of the microbial communities throughout the tract [13].

In healthy individuals, bacteria of the microbiome serve many beneficial functions [14]. The majority of the investigations into specific bacterial–host interactions have been conducted in intestinal epithelia, where the microbiome has been shown to play a critical role in normal immune defense. Briefly, a low-level inflammatory response is physiologic in the healthy intestine, which is initiated by commensal bacteria activating Toll-like receptors (TLRs) [15]. Local commensal bacteria adhere to epithelial cells and activate a variety of downstream inflammatory systems via interactions with immune cells including dendritic cells and Th17 cells [16]. Some commensal bacteria even appear to prevent higher-level inflammation and secrete products that actively relieve inflammation [17]. Bacteria also modulate tissue dynamics, as administration of *Bifidobacteria* has been shown to decrease intestinal permeability [18]. Although there is a paucity of parallel mechanistic investigations within the head and neck regions, it is likely that the microbiome similarly contributes to healthy baseline function.

## 3. The Microbiome May Be Altered in HNSCC

Bacteria have been linked to several kinds of cancer, notably *Helicobacter pylori (H. pylori)* with gastric cancer and *Fusobacterium nucleatum (F. nucleatum)* with colorectal cancer [19]. Emerging evidence within the past decade has suggested altered microbiomic content in patients with cancers of the head and neck compared to healthy controls. However, the specific changes remain unclear.

While many studies suggest that the bacteria of the microbiome associated with HNSCC is altered compared to healthy tissue, it is unclear whether the microbiome is directly involved in disease pathogenesis or simply a marker that reflects an altered tissue environment. Figure 1 illustrates the current understanding of how microbial infection contributes to HNSCC pathogenesis. One hypothesized pathogenic role for the microbiome is that specific bacteria may contribute to inflammation that promotes carcinogenesis [20,21]. Known HNSCC risk factors, including tobacco, alcohol, and poor oral hygiene, may facilitate this inflammatory state via the breakdown of tissue integrity and resultant exposure of TLR’s to the microbiome [6]. In addition to promoting a pro-inflammatory state, prior studies suggest that periodontal pathogens such as *Porphyromonas gingivalis* (*P. gingivalis*) may promote immune evasion by inducing expression of programmed death ligand (PD-L1/B7-H1) in squamous cell cancer cells [22,23]. Other studies have suggested that bacteria such as *F. nucleatum* are associated with tumor suppressor hypermethylation, resulting in induction of cell proliferation, as well as impaired transcriptomic regulation of immunomodulatory genes (i.e., latexin) [19]. There is also evidence that the microbiome is altered in precancerous states, further emphasizing a potential stepwise and mechanistic role for microbiota [24]. Conversely, the biomarker theory of the altered microbiome suggests that the cancer itself may alter the tissue environment such that different bacterial specimens are allowed to thrive [5]. It is, of course, possible that each of these hypotheses has validity, and a rigorous understanding of changes in the microbiome may enable earlier detection of disease and improved risk stratification [24].

There is a lack of consensus concerning the salient changes to the microbiome that are seen in patients with HNSCC. To explore the discrepancies, we performed a topical literature review of works in PubMed published between 2012 and 2023 investigating the microbiome of human subjects with HNSCC through clinical specimen collections (tissue, swab, saliva, or some combination). All studies reviewed are represented in Appendix A.

Many studies directly compared the microbiomes of samples from patients with HNSCC to samples from healthy controls (Table 1). Others investigated whether specific microbiome profiles were associated with either risk of clinical outcomes (including cancer development, mucositis, metastases, or survival) or genetic conditions (i.e., specific mutations, and other neurodegenerative diseases) (Table 2).

Our findings concurred with the predicted lack of scientific agreement regarding bacterial diversity in cancerous specimens (Table 1 and Table 2). Some studies report increased bacterial diversity in cancerous tissue compared to normal tissue [25,26,27,28], while others repeat either decreased diversity [29,30,31,32] or no significant differences at all [17,33,34,35]. One study found differences in diversity even between different cancer mutation subgroups [36]. Furthermore, another group elucidated diversity alterations when comparing the microbiome of cancerous tissue to adjacent normal tissue within the same subject, but no differences in diversity when comparing HNSCC patients to healthy controls at the cohort level, suggesting that alterations in the microbiome may be patient specific [19].

The discrepancies in findings may be, at least in part, due to diverse experimental designs and sample collection techniques, different sequencing practices, and bias introduced in statistical analysis (Appendix A). While some studies utilize tumor biopsies, investigations utilizing saliva samples may suffer from the limitation that they combine contents from multiple anatomic subsites with baseline differences in microbiomes and may not reflect the microbiome of the tumor subsite, while mouthwash samples may be limited by alterations in the microbiome caused by exposure to the mouthwash, itself [33]. In addition, many of the studies included in this review utilize 16s rRNA sequencing to detect microbial communities; however, this technique can carry the risk of introducing bias in taxa identification due to differential amplification of certain taxa in the PCR process, even with well-defined primers [37]. The size of the library (the sequencing depth) and statistical processing of data can also introduce bias into data analysis, as fundamentally rare taxa may be missed in studies with low sequencing depth. As such, microbiome data can be hugely impacted by a variety of components involved in its generation and analysis—thus, differences in such components across studies may be the source of discordance on the influence of the microbiome on HNSCC pathogenesis. Furthermore, the general skew of investigations into OSCC over other subsites may both reflect a sampling bias in terms of ease of prospective collection of saliva/oral cavity samples as well as a consequence of the increased prevalence of OSCC cancers [38].

Beyond microbial diversity, researchers have also sought to identify specific bacteria that are either enriched or depleted in cancerous tissue compared to normal (Table 1). Several studies demonstrated higher expression of *Lactobacillus* in HNSCC microbiomes compared to healthy controls [21,39,40]. Interestingly, *Lactobacillus* colonization in the colon has been proposed to prevent colorectal cancer via exertion of pro-apoptotic effects, suggesting diverse roles for the same bacteria within different areas of the body [41]. In addition, comparing cancerous tissue to normal, several groups found an increased presence of *Fusobaceteria* [42,43], and multiple investigations demonstrated a decrease in *Streptococcus* [5,19,28,44]. Studies have also suggested the association of specific microbial signatures with disease states, including precancerous lesions and risk of metastases (Table 2) [24,43]. There remains a need to further prospectively characterize clinical outcomes based on microbiome in both HNSCC patients as well as asymptomatic, at-risk patients.

There may also be elements of the microbiome that protect against HNSCC. In a large case–control study, Hayes et al., posited that higher concentrations of *Corynebacterium* and *Kingella* were associated with a decreased risk of HNSCC, possibly due to the carcinogen metabolic abilities of these bacteria [33].

HPV-positive and HPV-negative carcinomas are thought to potentially be clinically different disease entities [4]. Many microbiome studies assess both HPV-positive and -negative patients (see Appendix A); a subset investigated based solely on HPV status, with several addressing HPV-negative carcinomas [17,19,44] and fewer focusing on HPV-positive HNSCC patients [35,40]. Many studies do not report the HPV status of their subjects [5,6,27,29,32,34,45,46]. While there is a well-characterized mechanistic link between HPV status and the development of squamous cell carcinoma, these studies suggest that the microbiome may affect this relationship in a manner yet to be elucidated.

Together, these studies (Table 1 and Table 2, Appendix A) have demonstrated many different associations between bacterial microbial signatures and HNSCC. This underscores the need to understand the mechanistic links at play and whether therapies targeting the microbiome may have a role in the treatment of HNSCC.

**Table 1 cancers-15-05810-t001:** Topical summary of the literature investigating oral microbiome in HNSCC subjects.

Sample	Citation Number	Technique	Bacterial Diversity		Notable Specific Bacterial Populations Enriched or Depleted in Cancer	
			Alpha	Beta	Enriched	Depleted
Tissue	[20]	16S rRNAseq	NR		NR	*Entereobacteriaceae*
	[25]	16S rRNAseq	Increased	NR	*Fusobacterium*, *Capnocytophaga*, *Alloprevotella*	*Streptococcus*, *Veillonella*, *Lautropia*
	[5]	16S rRNAseq	Increased	+	*Fusobacterium*, *Prevotella*, *Capnocytophaga*	*Streptococcus*, *Veillonella*, *Parvimonas*
	[17]	16s rDNAseq	No difference	−	*Parvimonas*	*Actinomyces*
	[27]	16S rRNAseq	Increased	+	*Fusobacterium*, *Prevotella*, *Porphyromonas*	*Streptococcus*, *Veillonella*, *Rothia*
	[29]	16S rRNAseq	Decreased	+	*Stenotrophomonas*, *Ruminococcus*, *Comamonadaceae*	*Tannerella*, *Veillonella*, *Kingella*
	[40]	PathoChip	NR		*Lactobacillus*, *Lactococcus*, *Proteus*	NR
	[35]	16S rRNAseq	No difference	+	*Firmicutes*, *Actinobacteria*	*Spirochetes*, *Synergistetes*, *Fusobacteria*
	[47]	16s rRNAseq	NR	+	*Fusobacteria* and *Spirochaetes*	*Firmicutes* and *Actinobacteria*
Saliva	[39]	16S rRNAseq	Increased	+	*Lactobacillus*, *Streptococcus*, *Staphylococcus*	*Aggregatibacter*, *Haemophillus*, *Neisseria*
	[21]	16S rRNAseq	Increased	+	*Streptococcus*, *Lactobacillus*, *parvmonas*	*Leptotrichia*, *Fusobacterium*
	[33]	16S rRNAseq	No difference	-	*Actinobacteria*	NR
	[42]	RNASeq	NR	-	*Fusobacteria*, *Selenomonas*, *Capnocytophaga*	NR
	[26]	16S rRNAseq	Increased	+	*Fusobacterium *	*Streptococcus*, *Haemophilus*, *Porphyromonas*
	[6]	16S rRNAseq	NR		*Prevotella*, *Fusobacterium *	*Streptococcus*
	[34]	16S rRNAseq	No difference	+	None	None
	[44]	16S rRNAseq	No difference	+	*Fusobacterium*, *Prevotella*, *Alloprevotella*	*Streptococcus*
	[31]	16S rRNAseq	Decreased	+	*Streptococcus*, *Gemella*, *Veillonella*	*Haemophilus*, *Veillonella*, *Fusobacterium*
	[48]	16S rRNAseq	No difference	+	*Fusobacteria*, *Prevotella*, *Veillonella*	*Neisseria*, *Rothia*, *Rhodotorula*
	[49]	16S rDNAseq	NR		*Granulicatella*, *Alloscardovia*, *Stenotrophomonas*	*Moryella*, *Kingella*, *Centipeda*
	[30]	16S rRNAseq	Decreased	+	*Lactobacillus*, *Ochrobactrum*, *Parvimonas*	*Neisseria* and *Phyllobacterium*
	[50]	16S rRNAseq	No difference	+	*Lachnospiraceae*, *Eikenella*	*Lactobacillus*, *Bacillus*, *Bifidobacteriaceae*
Tissue/ Saliva	[32]	16S rRNAseq	Decreased	+	Tissue: *Acinetobacter*, *Fusobacterium*, *Campylobacter* Saliva: *Streptococcus*, *Prevotella*	NR
	[19]	16S rRNAseq	Decreased	+	Tissue: *Fusobacterium*, *Peptostreptococcus*, *Johnsonsella* Saliva: *Fusobacterium*, *Alloprevotella*, *Prevotella*	Tissue: *Streptoccocus*, *Neisseria*, *Veillonella* Saliva: *Streptoccocus*, *Neisseria*, *Rothia*
	[43]	16S rDNAseq	Decreased	+	*Fusobacterium*, *Peptostreptococcus*, *Prevotella*	*Streptococcus*, *Neisseria*, *Haemophilus*
	[51]	16S rDNAseq	Increased *	+	*Fusobacterium*, *Prevotella*, *Actinomyces*	*Streptococcus*, *Veillonella*, *Rothia*
	[52]	16S rDNAseq	Increased	+/− **	*Veillonella*, *Fusobacterium*	*Streptococcus*, *Neisseria*, *Prevotella*
Swab	[28]	16S rRNAseq	Increased	+	*Fusobacterium*, *Peptostreptococcus*, *Prevotella *	*Streptococcus*

+ = significant difference in bacterial colonies between HNSCC and control. − = no significant difference in bacterial colonies between HNSCC and control. * Higher abundance of known oral pathogens. ** See Appendix A for details.

**Table 2 cancers-15-05810-t002:** Investigating the associations between distinct microbiome profiles and the risk of clinical outcomes or genetic conditions.

Sample	Citation Number	Technique	Stratification	Bacterial Diversity		Notable Specific Bacterial Populations Enriched or Depleted	Key Findings
				Alpha	Beta		
Tissue	[53]	16S rRNAseq	Chemotherapy induction	Decrease	+	Enriched in induced chemotherapy group: *Mycoplasma* and unidentified *Veilloneliaceae*. Depleted in induced chemotherapy group: *Veillonella*, *Rhodococcus* and *Acinetobacter*.	In a non-induced chemotherapy group, *Fusobacterium* and *Actinomyces* were associated with more advanced stage of disease.
	[54]	16S rRNAseq	Length of survival	Decreased diversity in patients who survived less than 3 years compared with those who survived greater than 3 years	Significant difference in bacterial communities between those who survived less than 3 years and those who survived greater than 3 years.	Enriched in cases with survival less than 3 years: *Methyloversatilis* and *Schlegelella*. Enriched in cases with survival greater than 3 years: *Bacillus*, *Lactobacillus* and *Sphingomonas*.	Patients with tumors with increased dysbiosis exhibited shorter overall survival than those with less dysbiosis.
	[55]	RNAseq	Subsite Location	NR		Enriched Oral HNSCC: *Fusobacterium*, *Leptotrichia*, *Selenomonas* and *Treponema*. Enriched Nonoral HNSCC: *Clostridium* and *Pseudoalteromonas*.	Microbial signatures were correlated with the Kyoto Encyclopedia of Genes and Genomes pathways for both oral and non-oral cancers. Oral cancers showed signatures involved in neurodegenerative diseases and non-oral cancers showed signatures involved in HSV-1 infection.
	[56]	RNAseq		NR		Enriched by Subsite Oral Cavity: *Pseudomonas* Oropharynx: *Actinomyces* and *Sulfurimonas* Larynx: *Filifactor*, *Pseudomonas*, and *Actinomyces *	Microbial diversity was dependent on tumor location (oral cavity, oropharynx, larynx).
Saliva	[36]	16S rRNAseq	Tumor Mutational Characterization	Significant difference between mutational signal cluster cluster 2 and 3	Slight difference in bacterial communities between mutational signal cluster 1, 2 and 3	Enriched in MSC1 and MSC2: *Rothia* Enriched in MSC2: *Firmicutes* Enriched in MSC2 and MSC3: *Selenomonas* Enriched in MSC3: *Capnocytophaga*	Inferred functional assessment for microbial communities across mutational states showed differential enrichment in pathways linked to cell-mobility
	[57]	16S rRNAseq	Development of Oral Mucositis	No difference *	− *	See key findings	Patients with enrichment in *Cardiobacterium*, *Granulicatella*, *Prevotella*, and *Fusobacterium* had increased risk of developing early onset severe oral mucositis after chemoradiation. Patients with enrichment of Streptococcus had a decreased risk of developing early onset severe oral mucositis after chemoradiation.
	[45]	16S rRNAseq	Metastasis	No difference between metastatic cancer and non-metastatic cancer	Significant difference between metastatic cancer and non-metastatic cancer	Enriched in metastatic group: *Prevotella*, *Stomatobaculum*, *Bifidobacterium*, *Peptostreptococcaceae*, *Shuttleworthia* and *Finegoldia* Enriched in non-metastatic group: *Neisseria*, *Haemophilus*.	A machine learning program used the oral microbiome to predict lymph node metastases with 86.3% accuracy.

+ = significant difference in bacterial colonies between HNSCC and control. − = no significant difference in bacterial colonies between HNSCC and control. * From baseline study cohort before treatment with radiation/chemotherapy.

## 4. Animal Models Suggest the Microbiome May Contribute to the Development and Progression of HNSCC

Most in vivo studies of the HNSCC microbiome have utilized the 4-nitroquinolone-1-oxide (4-NQO) mouse model. 4-NQO, which is a water-soluble quinolone derivative, reliably produces premalignant and malignant lesions when spread on the palatal surface of mice [58]. These lesions have been shown to both histologically and morphologically resemble HNSCC [59]. The carcinogenic effect is largely due to an interaction between 4-NQO and the nucleophilic part of DNA, resulting in the formation of DNA adducts around guanine residues [60]. Common models of HNSCC include the administration of 4-NQO in the drinking water of mice for anywhere from 8 to 25 weeks, followed by sample collection [61]. Tumor samples can be further processed to assess for any transcriptional, proteomic, or histological abnormalities [61]. Such methods provide a robust platform to understand the development of HNSCC and interrogate its associated risk factors.

Some studies have used the 4-NQO model to explore the impact of the microbiome on HNSCC (Figure 2). Binder-Gallimidi et al., found that administration of *P. gingivalis* and *F. nucleatum* contributed to carcinogenesis in mice treated with the oral carcinogen 4-NQO, with evidence demonstrating increased activity on the TLR2/TLR4/IL-6/STAT3 pathway [8]. Furthermore, coinfection was also associated with tumor progression: mice treated with *P. gingivalis* and *F. nucleatum* in addition to 4-NQO developed larger and more invasive tumors relative to mice administered 4-NQO alone [8]. Frank et al. demonstrated that antibiotic depletion of the mouse microbiome delayed oral tumorigenesis in mice administered 4-NQO, and transplantation of microbiota from mice with cancer accelerated it [30]. Furthermore, Stashenko et al. showed that gnotobiotic mice inoculated with the microbiome of both healthy and OSCC-positive mice experienced increased tumor sizes and numbers when exposed to 4-NQO compared to microbiome-free mice, implying further nuance in bacterial–tissue dynamics [62].

These studies suggest that certain bacterial populations may contribute to the development of cancer; however, mice and humans have notably different bacterial populations and immune system compositions [63,64]. This limitation of animal models, along with the time and cost associated with animal studies, underscores the need for reliable in vitro systems to model the interaction between human cancer cells and microbes. While OSCC predominates the clinical literature surrounding the microbiome of the head and neck (Appendix A), this is even more true for in vitro studies. As such, the remainder of this review will focus on experiments that have modeled host–microbe dynamics within the oral cavity, with the understanding that additional investigations are required to continue these inquiries within other regions of the head and neck. 

## 5. In Vitro Modeling of Host–Microbe Interactions in HNSCC

Conventional cell culture models of host–microbe interactions typically entail co-culturing a confluent monolayer of epithelial cells with single or multiple microbial species (Figure 3A,B) [65]. Some studies additionally incorporate biofilms to capture the unique properties of polymicrobial plaque formation (Figure 3C) [66,67]. Numerous studies have co-cultured immortalized gingival epithelial cells with *P. gingivalis* to investigate the potential impact of infection on the malignant transformation of host epithelial cells. Exposure of human oral epithelial cells to *P. gingivalis* promoted cell proliferation [68,69], inhibition of pro-apoptotic pathways [70,71], and enhanced both cell migration and invasion [68,69,72,73]. Additionally, Sztukowska et al., demonstrated that *P. gingivalis* co-cultured with oral cancer cells promoted the epithelial-to-mesenchymal transition (EMT) through upregulation of Zeb1, a canonical EMT transcription factor. These co-cultured oral cancer cells were found to have increased expression, secretion, and activation of pro-matrix metalloproteinase-9 (MMP-9), facilitating increased invasiveness [74]. Groeger et al. demonstrated that *P. gingivalis* co-culture led to the upregulation of B7-H1 and B7-DC receptors in squamous carcinoma cells, further implicating a role of bacterial infection in facilitating immune evasion by oral cancer cells [75].

Binder-Gallimindi et al. demonstrated that co-culture of *F. nucleatum* with two human OSCC lines increased IL-6 expression and induced key molecular markers, such as cyclin D1, MMP-9, and TNFα, which are hypothesized to be involved in oral cancer cell invasion and tumor aggressiveness [8]. Furthermore, they found that exposure to these pathogens stimulated cellular proliferation [8]. These findings were supported by Harrandah et al., who demonstrated that exposure of human oral cancer cells to *F. nucleatum* led to a significant increase in STAT3 and MYC, promoted EMT through increased expression of TGF-β, ZEB1, MMP-9, and MMP-1, and enhanced invasiveness in vitro [72].

However, while monolayer culture systems are easily accessible and low cost, they do have several limitations. The biggest is the inability of 2D cell lines to adequately capture inter- and intratumoral cancer cell heterogeneity [76], which is critical in modeling host–microbe interactions given patient- and subsite-specific differences in microbial profiles associated with HNSCC [55]. In monolayer culture, there is a loss of the cell–cell and cell–extracellular interactions that are observed in vivo and that have been shown to play important roles in response to stimuli [77,78,79,80]. Three-dimensional culture systems, including organotypic culture and 3D organoids, may address some of these concerns by better recapitulating intercellular interactions and differentiation gradients [81] and, as such, enable more physiologic modeling of host–microbe interactions.

## 6. Organotypic 3D Culture Systems to Study Host–Microbe Interactions

Organotypic 3D culture models have largely focused on *P. gingivalis* and *F. nucleatum* as bacterial targets of choice for experimentation. As findings from prior 2D monolayer co-culture experiments implicated a role of these bacterial pathogens in basement membrane dysregulation; similar results were found employing organotypic 3D culture, as with Andrian et al., who showed the upregulation of these same MMPs upon co-culturing of human oral mucosa with *P. gingivalis* [82]. In addition, this modeling system has been shown to capture the unique immune microenvironment following microbe co-culture of the oral epithelium. *P. gingivalis* co-culture was also shown to initiate a pro-inflammatory response in the oral epithelium, increasing expression of IL-1B, IL-6, IL-8 and TNF-a [83]. As such, this model has served as a valuable resource to study the impact of single-species infection on tissue dynamics.

Some studies have also used the organotypic 3D model to explore the impact of synthetic biofilms on the oral epithelium. Gursoy et al., found that exposure of an organotypic dento-epithelial model to biofilms of two different strains of *F. nucleatum* resulted in the expression of the antimicrobial peptides, human B- defensin-2 and -3, as well as cathelicidin—a response known to occur within the gingival epithelium in vivo [84]. Other groups have focused on using biofilms of multiple microbial species to model complex host–microbiome interactions. In particular, Brown et al., exposed co-cultured immune cells and oral mucosa to biofilms containing up to ten common oral microbes and found upregulation of the pro-inflammatory cytokine CXCL10 [85]. Overall, this modeling system demonstrates a capability to examine the theorized pro-inflammatory influence of microbial/polymicrobial infection, as well as the ensuing dysregulation of the basement membrane on malignant transformation in HNSCC.

## 7. Three-Dimensional Organoids to Model Host–Microbe Interactions

Organoids are single-cell-derived 3D clusters of epithelial cells grown in a basement membrane gel [86]. These cultures can be derived from adult stem cells present in a variety of tissue sampling methods including excess tissue from surgical resections, solid needle or punch biopsies, and even fresh frozen tissue samples [87]. HNSCC 3D organoids have been shown to recapitulate the morphologic characteristics and response to stimuli of the parent tissue from which they were derived [88], making them compelling models to study host–microbe interactions in disease development. Though 3D organoid culture systems have been used to model host–microbe interactions at distal sites of the gastrointestinal tract (i.e., stomach, small intestine, colon) there is a paucity of the literature examining this interaction in the context of HNSCC. In 3D organoid culture, it is the “outer” basal layer that is typically exposed to stimuli such as growth factors in media and, in studies of host–microbe interactions, the co-cultured microbes. In reality, however, host–microbe interactions happen at the apical mucosal surface which corresponds to the center of the organoid. To this end, there have been several co-culturing approaches described (Figure 4).

### 7.1. Co-Culture via Microinjection

The first technique includes microinjection of microbes into the lumen of intact organoids. Using this approach, microbes are introduced to the apical aspect of the polarized epithelium, achieving an accurate and efficient co-culture method with limited cytotoxicity [89,90,91,92]. An advantage to this approach is the freedom to optimize a precise dose of microbes to be injected. Such optimized dosages can be administered at a single time to be measured and characterized over time [89]. This approach has been used to investigate the antibacterial function of defense secreted by epithelial cells and can be readily translated to the investigation of additional products [89]. Microinjection represents a useful approach to examining host–microbe interaction for high-throughput applications. For instance, Williamson et al., developed and validated a high-throughput semi-automated organoid microinjection system for both cargo delivery and lumen sampling of colonic organoids [93]. Despite these advantages, there are associated limitations to the microinjection approach. This approach is optimized to microinject a single bacterial species, which may not adequately demonstrate the dynamic interplay between multiple microbial species in the microenvironment. Furthermore, the optimization of oxygen and nutrient levels to maintain symbiotic microbe–host interactions is still in its early stages [94]. To preserve microinjection accuracy and mitigate damage to organoid cells with consequential leakage of injected bacteria towards the basolateral side, this approach requires specialized equipment with the appropriate setup, as well as adequately trained users [93].

### 7.2. Microbial Infection of Dissociated Epithelial Cells

In this approach, organoids are dissociated into a single-cell suspension and subsequently co-plated with microbes/microbe products [95,96,97,98]. As the organoids begin to form within the extracellular matrix, the co-cultured microbes become incorporated into the apical surface of the organoid to model infection. Unlike the microinjection approach, this method is relatively simple and does not require special equipment to perform. In their study, Huang et al., adopted this approach to perform a quantitative proteomic analysis comparing total proteomes of intestinal organoids co-cultured with *Listeria monocytogenes*. Their analysis found over three hundred differentially expressed proteins, which were related to host immune response, biological metabolism, and energy metabolism [99]. A drawback to this approach is that it is challenging to pinpoint the initial time of interaction between host epithelial cells and microbes, and the infection efficiency is variable depending on the microorganisms utilized. As this approach leads to bacterial interactions with both the apical and basolateral sides of the organoids, the physiological relevance can be questioned. Additionally, it is challenging to quantify the amount of microbe entrapped within the organoid’s lumen [90,100].

### 7.3. Epithelial-Microbial Co-Culture Using Organoid-Derived Monolayers

As an alternative approach, organoids can be dissociated into single cells and subsequently in a 2D monolayer. Previous studies have shown that organoid-derived monolayers exhibit differentiation that reflects the parent tissue from which it was derived [101,102]. Nickerson et al. demonstrated the presence of multiple cell types, such as enterocytes, mucus-producing goblet cells, and M cells were present in their human intestinal organoid-derived epithelial model (HIODEM) [102]. Thorne et al. demonstrated similar findings, showing that intestinal organoid-derived monolayers generated major intestinal cell types, and additionally organized into proliferative and differentiated zones [101]. When the single-cell suspension is seeded onto a coated plate with an extracellular matrix-based hydrogel such as Matrigel (Corning, Glendale, AZ, USA) or collagen, the cells adhere to the surface with the apical side facing upward. Directly adding microorganisms to culture media in this orientation facilitates microbe-epithelial contact [101]. Due to the simplicity of this approach, there have been a growing number of studies using this technique to examine the characteristics of microbial-epithelial interactions including microbe-mediated immune activation and cytokine responses to pro-inflammatory stimuli [103,104,105]. This paradigm may be useful for co-culture experiments involving aerobic and facultative anaerobic bacteria since the coated plate has a larger area in contact with oxygen. However, co-culturing obligate anaerobic bacteria with epithelial cells in this system may be challenging due to differential oxygen requirements [106,107,108]. It is also critical to note that as organoids transition to monolayer culture, there is an appreciable loss of cellular heterogeneity and intercellular interactions, and this may pose a challenge to recapitulate the tissue architecture and function observed in vivo [100].

### 7.4. Three-Dimensional Organoids to Examine Microbial Contributions to Carcinogenesis in the Aerodigestive Tract

To our knowledge, there has only been one study conducted involving the use of organoids to study host–microbe interactions within the oral cavity. Bugueno et al., utilized a 3D spheroid model of the gingiva to better understand interactions between different cell types and the impact of *P. gingivalis* infection [109]. Elsewhere along the GI tract, however, there have been studies utilizing organoids to investigate the interactions between enteric microbes and epithelial cells [89]. For example, Zhang et al., demonstrated that intestinal organoids co-cultured with *S. enterica* revealed bacterial adherence and invasion of intestinal epithelial cells, and elicited NF-kappaB signaling activation [98]. Bartfeld et al., observed a similar effect modeling *H. pylori* infection of gastric organoids, demonstrating that microinjected bacteria became tightly associated with the organoid epithelium and induced expression of NF-kappaB target genes including IL-8 [110]. In addition, to study the role of *H. pylori* infection on gastric carcinogenesis, Wroblewski et al. successfully infected murine gastric organoids with *H. Pylori* cagA+ wild-type via microinjection, which resulted in increased epithelial cell proliferation and B-catenin nuclear translocation [111]. These findings have since been validated using human gastric organoids [112,113,114], demonstrating the use of this 3D model system as a tool for both discovery and validation. Insights gleaned from such studies have added to our understanding of microbial mechanisms of infection, host–microbe crosstalk, as well as patterns of physiological response to microbial infection [115].

## 8. Future Directions: 3D Microbiome Co-Culture Models to Investigate HNSCC Carcinogenesis

Given recent advancements in co-culturing techniques, experiments employing the use of 3D organoids to study the microbiomic influences on HNSCC may provide vital insights into the complex pathogenesis of this disease, as well as the role of bacterial infection in the progression from normal mucosa to precancerous lesions to HNSCC. The impact of the microbiome on neoplastic progression can be evaluated by co-culturing 3D organoids derived from normal oral mucosa with specific microbes of interest while assessing the development of cellular atypia, patterns of invasion, and molecular changes as readouts of progression. Furthermore, co-culturing human HNSCC organoids with microbes would offer insights into the influence of microbial infection on the expression heterogeneity in HNSCC. For instance, *F. nucleatum* has previously been shown to induce EMT, a known risk factor in metastatic disease [116], in oral epithelial cell lines [117], providing the basis for future studies to focus on characterizing EMT within co-treated 3D organoids. Our group has shown that HNSCC tumors express a partial-EMT (p-EMT) program, which retains certain epithelial characteristics and has been shown to drive tumor heterogeneity and invasion [118]. As such, inquiry into the relationship between *F. nucleatum* and p-EMT may lend valuable information to the underlying influence of the microbiome on carcinogenesis.

Additionally, 3D organoids may also present a unique opportunity to explore the viral influences on HNSCC development. There is a clear relationship between HPV infection and the development of oropharyngeal HNSCC [119]. While the prognosis of HPV-positive HNSCC is better than that of HPV-negative [119], much remains unknown regarding the potential pathogenic involvement of the microbiome [35,40]. Some studies have suggested a mechanistic interaction between HPV and the microbiome in other disease processes such as bacterial vaginosis [120]. However, such an interaction has not been similarly discovered in HNSCC. Previous studies validated the ability to productively infect HNSCC organoids with HPV [88]. As such, co-cultured HPV-positive HNSCC organoids with candidate microbes may serve as an effective method for uncovering mechanisms of disease that influence patient morbidity and mortality.

There are limitations associated with the use of 3D organoids. Although 3D organoids can be used to study interactions between microbes and epithelium, immune and stromal cells are not represented in organoid culture. While some immune and stromal cells are retained during early passages, these components are eventually lost in culture [121]. Given the well-described role of oral microbial infection in the relationship between chronic inflammation and cancer, the incorporation of immune cells into in vitro experiments may be useful in improving our understanding of this link. Though previous studies have demonstrated alterations in the expression of pro-inflammatory mediators such as IL-6, IL-8, and TNF-a in epithelial cells co-cultured with microbes, experimental approaches involving direct co-culture with organoid and immune cells remain in nascent stages [66,122]. Additionally, there are several approaches to co-culturing epithelial organoids with microbes, each with its advantages and disadvantages, and it is unclear which method best recapitulates physiological responses to exposure. 

## 9. Conclusions

The microbiome is an important component of human physiology in health and disease. Despite growing interest in this area of study over the past decade, our understanding of microbe-epithelial interactions in HNSCC remains limited. Animal models and conventional in vitro studies have made significant strides in examining the role of the microbiome in HNSCC, from highlighting candidate microbes and patterns of microbiome composition implicated in disease to developing theorized mechanisms of pathogenesis. However, these modeling systems are met with their challenges. Three-dimensional culture techniques serve as a promising tool to investigate the complex interplay between the microbiome and the development of HNSCC while faithfully recapitulating human features in a manner that animal models and conventional in vitro models lack. Insights gleaned from such studies may aid strategies to improve risk stratification, an underlying issue contributing to the continued poor prognosis in HPV-negative HNSCC, and potentially guide treatment.

## Figures and Tables

**Figure 1 cancers-15-05810-f001:**
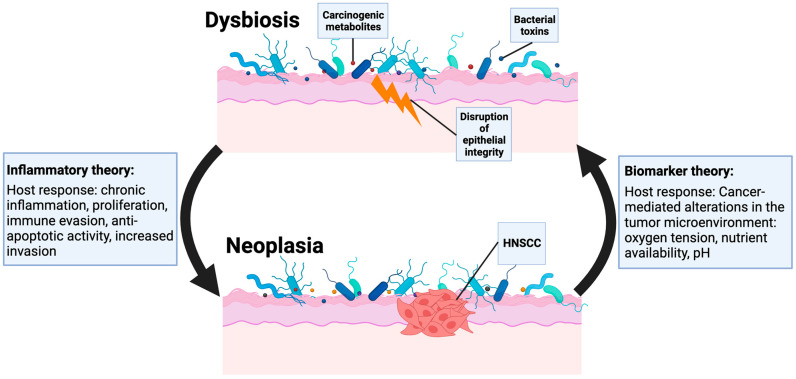
Schematic illustration of the current understanding of the relationship between microbial infection and carcinogenesis. This figure was created using Biorender.com (accessed on 23 October 2023).

**Figure 2 cancers-15-05810-f002:**
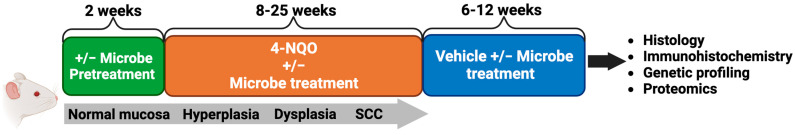
Schematic of 4-NQO model to study microbial infection in HNSCC. Mice inoculated with microbes of interest preceding (green rectangle) or in conjunction with 4-NQO administration (orange rectangle), followed by a period of vehicle administration with or without microbial treatment (blue rectangle). Downstream analysis upon tissue collection includes histology, immunohistochemistry, genetic profiling, and proteomics. This figure was created using Biorender.com (accessed on 23 October 2023).

**Figure 3 cancers-15-05810-f003:**
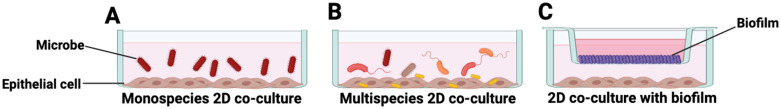
Standard 2D monolayer culture systems studying host–microbe interactions. (**A**) Two-dimensional monolayer culture with single-species microbes. (**B**) Two-dimensional monolayer co-culture with multispecies microbes. (**C**) Two-dimensional monolayer culture with biofilm suspended on a Transwell insert. This figure was created using Biorender.com (accessed on 23 October 2023).

**Figure 4 cancers-15-05810-f004:**
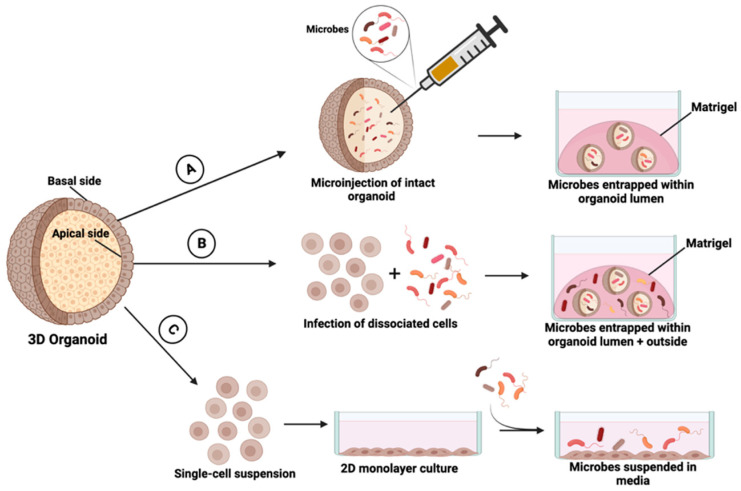
Utilization of adult stem cell-derived 3D organoids to examine host–microbe interactions. (**A**) Microbial infection of intact organoid via microinjection method. (**B**) Microbial infection of dissociated epithelial cells prior to organoid development. (**C**) Microbial infection in organoid-derived monolayer culture. This figure was created using Biorender.com (accessed on 23 October 2023).

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
