# Peer review of "Experimental Modeling of Host–Bacterial Interactions in Head and Neck Squamous Cell Carcinoma"

_cancers, 2023, doi:10.3390/cancers15245810_

Round 1

Reviewer 1 Report (Previous Reviewer 2)

Comments and Suggestions for Authors

Authors have answered questions and significantly improved the manuscript.

Reviewer 2 Report (Previous Reviewer 3)

Comments and Suggestions for Authors

The authors have fully answered my previous suggestions, and  the manuscript has been significantly improved.

This manuscript is a resubmission of an earlier submission. The following is a list of the peer review reports and author responses from that submission.

Round 1

Reviewer 1 Report

Comments and Suggestions for Authors

Okolo, Honzel, et al. present a narrative/topical review of in vitro modeling of host-microbiome interactions in head and neck squamous cell carcinoma (HNSCC). This is a rather large topic, and the authors should be commended on tackling it. There is some value to the current work; but, additional edits/clarifications are needed prior to publication.

- The title includes the phrase “in vitro” but the abstract states that the review will include “in vivo” modeling. And, there is indeed a small section on in vivo modeling. So—which is it? If the review is to include in vivo modeling; then, that section should be expanded and the title edited—with the phrase “in vitro” removed. Alternatively, remove the in vivo modeling section. That said, this reviewer would suggest keeping the in vivo modeling section, expanding it, and adding a figure to what is otherwise a rather text-heavy and bland looking review.

- The first paragraph of the introduction provides far too brief and generic an overview of HNSCC. For example, the phrase “imprecise clinicopathologic features including depth of invasion, lymphovascular invasion, and perineural invasion” is unsubstantiated by any citations; and, rather, flies in the face of much literature validating the clinical utility of these features (think D’Cruz et al. NEJM for depth of invasion). As such, this portion of the manuscript should be expanded with citations to relevant literature.

- There is a brief mention of HPV in the first introductory paragraph and later in the manuscript; but, this is not focused upon, which is a big issue in a review of the microbiome and HNSCC. The microbiome includes the totality of microbes (e.g. viruses, fungi, etc.) and not just bacteria. This review reads as though it is focused on the oral bacteriome and HPV-negative oral cavity SCC—and this is the biggest issue: it is not explicitly framed as such. There is just a sprinkling of HPV and Candida throughout. A true review on the microbiome and HNSCC would forcibly include sinonasal malignancies, the nasal microbiome (including fungi), and feature HPV prominently. It would review nasopharyngeal carcinoma, EBV, and cutaneous SCC. Someone not familiar with the field would be misled reading the current work. Significant edits and clarifications are need throughout the entire manuscript to either: 1) state/narrow the focus specifically to the oral bacteriome and HPV-negative (musical) oral cavity SCC or 2) to broaden the work to include all head and neck squamous cell carcinomas and their associated microbes.

- Supplementary Table 1 provides a good amount of value to the current work and indirectly addresses some of the issues above. Consider formatting in such a way as to be able to include it in the main manuscript.

Reviewer 2 Report

Comments and Suggestions for Authors

In this manuscript, Okolo et al. reviewed the in vivo and in vitro modeling methods of investigating the role of the microbiome in HNSCC. This is a growing field and may help to understand the complex relationship between the microbiome and HNSCC. Some concerns of this review should be resolved.  

1. Authors mentioned that “However, there remains controversy regarding specific microbiome profiles in HNSCC tissue”. Indeed, the methods of sample collection, sample numbers, the eating habits of patients, the methods to detect microbiome, and statistical analysis method have huge effects on the results of the microbiome associated with HNSCC. Therefore, authors should carefully evaluate the publications and filter the studies with obvious defects, then draw sound conclusions. In addition, authors should add a section to carefully discuss the limitations of current studies and how to improve the accuracy of the studies of the microbiome associated with HNSCC.

2. The in vitro models of host-microbe interactions in HNSCC, either 2-D or 3-D, almost completely ignored immune systems. Missing of immune microenvironment should be considered and discussed in detail, because bacteria are very strong stimulation to human immune systems.

3. The title is “In vitro modeling of host-microbe interactions in head and neck squamous cell carcinoma”. However, due to the limitation of in vitro model, the in vivo model should be emphasized and described in detail. In vivo model should be used to confirm the results from the in vitro studies.  

4. A figure of describing the current understanding of how bacteria contribute to HNSCC is required.

5. Figure 1. Intracellular infection of some bacteria should be considered.  

6. Some typing errors in most subtitles, for example:

Page 6: “7.D organoids to model host-microbe interactions.” “8.Co-culture via Microinjection”

Page 5: “6.Organotypic 3D culture systems to study host-microbe interactions”

7. Page 8: “NF-kb signaling activation” should be kappa.

Reviewer 3 Report

Comments and Suggestions for Authors

A good review about in vitro modeling of host-microbe interactions in head and neck squamous cell carcinoma (HNSCC). Some suggestions should be noticed as below,

1) Now emerging data shows that cancer is an ecological disease, it is proposed as a multidimensional spatiotemporal "unity of ecology and evolution" pathological ecosystem (https://www.thno.org/v13p1607.htm). Host-microbe interactions in cancer as a part of cancer ecology, that is cancer microecology. Such update should be added.

2) From above, if cancer is thought as an ecological disease, what kinds of the relationships between host-microbe interactions HNSCC are involved? Competition, predation, mutualism or others?

3) In Figure 2, cells in 3D organoids are driven from? E.g. HNSCC cell lines, patient tissues, CSCs or stem cells?

4) How about in vitro modeling of host-microbe interactions from the progression of normal epithelium-precancerous lesions-HNSCC.
